# Transient Receptor Potential Channels in the Epithelial-to-Mesenchymal Transition

**DOI:** 10.3390/ijms22158188

**Published:** 2021-07-30

**Authors:** Charlotte Van den Eynde, Katrien De Clercq, Joris Vriens

**Affiliations:** 1Laboratory of Endometrium, Endometriosis & Reproductive Medicine, Department of Development and Regeneration, KU Leuven, Herestraat 49 Box 611, 3000 Leuven, Belgium; Charlotte.vandeneynde@kuleuven.vib.be (C.V.d.E.); katrien.declercq@kuleuven.vib.be (K.D.C.); 2Laboratory of Ion Channel Research, Department of Cellular and Molecular Medicine, KU Leuven, VIB Center for Brain & Disease Research, Herestraat 49 Box 802, 3000 Leuven, Belgium

**Keywords:** TRP channels, epithelial-to-mesenchymal transition, fibrosis, cancer

## Abstract

The epithelial-to-mesenchymal transition (EMT) is a strictly regulated process that is indispensable for normal development, but it can result in fibrosis and cancer progression. It encompasses a complete alteration of the cellular transcriptomic profile, promoting the expression of genes involved in cellular migration, invasion and proliferation. Extracellular signaling factors driving the EMT process require secondary messengers to convey their effects to their targets. Due to its remarkable properties, calcium represents an ideal candidate to translate molecular messages from receptor to effector. Therefore, calcium-permeable ion channels that facilitate the influx of extracellular calcium into the cytosol can exert major influences on cellular phenotype. Transient receptor potential (TRP) channels represent a superfamily of non-selective cation channels that decode physical and chemical stimuli into cellular behavior. Their role as cellular sensors renders them interesting proteins to study in the context of phenotypic transitions, such as EMT. In this review, we elaborate on the current knowledge regarding TRP channel expression and activity in cellular phenotype and EMT.

## 1. Introduction

### 1.1. Cell Phenotypes Transitions and Calcium

Epithelial and mesenchymal cells represent two important cell populations in mammals with prominent different features. Due to their tight junctions and cohesive interactions, epithelial cells are perfectly suited to facilitate the formation of a polarized continuous cell layer. In contrast, mesenchymal cells are not polarized and lack cell–cell interactions, but they contain motile properties that allow them to invade neighboring tissues. Consequently, epithelial and mesenchymal cells adopt different functions in an organ, which is exemplified by specific transport epithelia, e.g., epithelia in the kidney and intestines, and by the structural role of the mesenchymal stroma.

While the cellular phenotypes in adult tissues are mostly well defined, the ability of cells to reversibly switch between distinct cellular phenotypes is essential for normal development and physiological functioning. Indeed, the epithelial-to-mesenchymal transition (EMT), defined as the process during which epithelial cells acquire mesenchymal features, is a prerequisite for several developmental and physiological processes. Based on the biological context, EMT processes can be categorized into three different types, all of which result in different functional consequences. Type I EMT is associated with the developmental processes of embryo implantation, embryogenesis and organ development. The induction of invasive properties does not occur. The resulting mesenchymal cells can also undergo the reciprocal mesenchymal-to-epithelial transition (MET) to form secondary epithelia. Type II EMT is induced by inflammation, and therefore represents a repair-associated event used to reconstruct tissues following injuries, such as wound healing. Attenuation of the inflammation reaction that is concomitant with tissue injury and regeneration will result in the cessation of type II EMT, but can adversely result in fibrosis. Type III EMT occurs in neoplastic cells, in which mutations in oncogenes and tumor suppressor genes culminate in the formation of carcinoma cells that acquire invasive properties, which in turn allow them to disseminate to ectopic locations. Metastasis of malignant cancer cells is enabled via extracellular signaling of growth factors (GF) and cytokines within the tumor microenvironment (TME). As such, tumor cells lose their cell–cell interactions and exhibit increased migratory behavior. Consequently, these cells will leave the primary tumor and travel to distant metastatic sites.

Both the translation of the extracellular signals into gene expression alterations and the execution of cellular behavior require a strict regulation of intracellular calcium levels. Therefore, adjustments in calcium signaling can have a drastic effect on the cellular phenotype. Indeed, several studies revealed essential roles for calcium signaling in EMT induction in both normal development and cancer progression. For instance, inhibiting calcium signaling is detrimental for normal fin development in Xenopus. Stimulation of calcium release rescues the EMT event that is indispensable for migration of cells into the fin matrix during its development [1]. Moreover, EMT induction in breast cancer cells is calcium-dependent, since intracellular calcium chelation blocks the induction of many EMT markers [2]. Additionally, a role for calcium ions has been described in EMT induction in both prostate and lung cancer cells [3,4]. A detailed overview of the altered calcium signaling in cancer cells was written by Stewart and colleagues [5].

### 1.2. TRP Channels

Calcium signals with specific spatio-temporal characteristics are generated by an intricate interplay between “calcium toolkit” proteins, comprising calcium-permeable ion channels, pumps, exchangers and calcium-binding proteins. In recent decades, lots of research has been conducted regarding the roles of transient receptor potential (TRP) channels in health and disease. TRP channels are a superfamily of membrane proteins functioning as non-selective cation channels that are involved in the detection of thermal, chemical and mechanical stimuli. In mammals, 28 different TRP members have been identified, which can be divided into six subfamilies based on sequence homology: TRPA, TRPV, TRPC, TRPM, TRPP and TRPML. Interestingly, TRP channels possess different cation selectivity, ranging from being calcium impermeable but sodium permeable, e.g., TRPM4 and TRPM5, to being selective for calcium but not sodium, e.g., TRPV5 and TRPV6. There is wide heterogeneity among the gating mechanisms of TRP channels, including activation by voltage, temperature and various endogenous or exogenous ligands. As such, the TRP superfamily contributes to a variety of physiological functions, such as vision, hearing, taste perception, thermosensation and cellular responses to different environmental stimuli [6]. In fact, TRP channels constitute the largest group of sensory ion channels involved in the detection of environmental stimuli, and have therefore drawn attention in the field of developmental and cancer research. Indeed, multiple studies have speculated on the role of TRP channels in tumor progression and have suggested TRP channels as excellent potential targets for the development of new cancer treatment therapies and potential biomarkers for disease prognosis [7].

During the last few decades, ourselves and others have observed very distinct expression profiles of TRP channels in epithelial and mesenchymal cells. As such, TRPV5 and TRPV6 are very well described as epithelial channels involved in the regulation of calcium (re)absorption across epithelial membranes [8]. TRPV4 channels are expressed in airway epithelial and urothelial cells and potentially regulate immune responses [9,10]. TRPM members observed in epithelial cells include TRPM6 [11], TRPM4 and TRPM7 [12,13,14]. In contrast, the expression levels of TRPV2, TRPC1 and TRPC4 are mainly associated with mesenchymal cellular phenotypes [12,13,15,16,17,18]. Given the strong associations of TRP channels with certain phenotypes, the EMT might result in a phenotypic switch of TRP channels that will confer specific features to the cell phenotype. 

EMT is accompanied by a shift in the cellular transcription profile, resulting in decreased expression of E-cadherin, cytokeratin and claudin, and increased expression of N-cadherin and vimentin (Figure 1). TGFβ1 is a major player in EMT induction, which acts by inducing the transcription of E-cadherin repressor genes Snail1 and slug, through the Smad/PI3K/ERK pathway. Apart from TGFβ1, activation of the tyrosine kinase receptors by growth factors such as FGF, HGF, IGF1, EGF and PDGF can induce EMT as well. Moreover, hypoxic conditions, as seen in proliferating tumors, can facilitate EMT, since HIF-1α was shown to activate Twist, an important transcription factor for the EMT [19]. Finally, mechanical stress, often caused by increased matrix stiffness in tumors, can also induce EMT [20,21]. Interestingly, these EMT inducers have all been described as TRP channel modulators. Recently, the effects of GF signaling—including EMT-inducing factors such as TGFβ1—on TRP channel expression and functioning were reviewed [22]. Moreover, hypoxic conditions can modulate TRP channel activity and consequently induce a cytosolic calcium influx [23,24]. Furthermore, the actions of several members of the TRP superfamily can be modulated by mechanical stress [25,26]. 

Thus, alterations in the extracellular environment or dysregulation of GF signaling, e.g., during cancer progression, could interfere with cellular TRP channel expression signatures, resulting in altered calcium homeostasis and the induction of transcriptomic profiles facilitating the EMT. Alternatively, the acquisition of the mesenchymal phenotype might implicate upregulation of TRP channels that are involved in cellular migration and invasion. Therefore, the expression of these channels might be necessary to maintain a cellular phenotype obtained via the process of EMT. In the next section, we further elaborate on the specific role of TRP channels in the EMT process (Table 1). Only TRP channels with clearly described roles in the EMT are included in this overview.

## 2. TRP Channels in the EMT

### 2.1. TRP Channels in Non-Pathological EMT Processes

As mentioned previously, type I and II EMTs are physiological processes that occur during development and tissue repair. It should be noted, however, that while the expression of TRP channels has clear associations with phenotypic switching during said processes, direct evidence of their involvement in EMT as such is still lacking.

During embryo implantation, trophoblast stem cells of the trophectoderm undergo EMT to facilitate endometrial invasion and appropriate placental formation [57]. Interestingly, a recent study highlighted the spatio-temporal expression pattern of TRP channels during murine placental development, suggesting possible involvement in this process [58]. Altered expression signatures might indeed originate from EMTs and could orchestrate endometrial invasion and migration required for proper placental formation. 

Furthermore, the endometrium itself is a remarkable tissue that undergoes repetitive cycles of breakdown and repair involving physiological phenotypic switching. Indeed, EMT and the reverse process, MET, are required for normal endometrial functioning. For instance, in humans, monthly preparation for embryo implantation includes spontaneous MET of endometrial stromal cell during a process called decidualization, thereby obtaining an epithelial-like secretory phenotype. In contrast, luminal endometrial epithelial cells undergo EMT in order to accommodate invading trophoblasts [59]. Interestingly, a very distinct TRP channel expression signature can be observed between endometrial stromal and epithelial cells [12,13,60]. This suggests that TRP channels might indeed play a role in defining the endometrial cell phenotype and facilitate normal endometrial functioning.

During embryo development, EMT plays a major role in the formation of the neuronal tube. Recently, TRPM6 and TRPM7 have been identified as novel regulators of mediolateral and radial intercalation during neuronal tube closure, mainly via Mg^2+^ signaling [61]. Altered Mg^2+^ concentrations will initiate signaling pathways that will induce cellular motility and cytoskeletal rearrangements. 

Wound healing encompasses re-epithelialization of the tissue, which is sustained by the conversion of sedentary cells towards a migratory phenotype via EMT. Several authors have reported a role for TRP channels in this re-epithelialization process [62,63,64]. For instance, wound closure was significantly delayed in TRPV3 KO mice compared to normal mice [62]. Therefore, albeit not proven, a potential role of TRP channels in EMT associated with tissue repair is expected. 

### 2.2. TRP Channels in Pathological EMT Processes 

#### 2.2.1. Cancer

Type III EMT occurs in neoplastic cancer cells, thereby facilitating the acquisition of mesenchymal properties, such as increased motility and invasion. Moreover, loss of cell–cell adhesion via E-cadherin repression further increases the potential of cancer cells to break free from the primary tumor and disseminate into the bloodstream, resulting in metastatic diseases. Several studies have defined a role for TRP channels in this type of EMT, most of which are related to cancer development. An overview of other plasma membrane ion channels and their involvement during EMT in cancer was presented in detail elsewhere [65]. A more detailed overview of TRP channels in cancer development, progression and their therapeutic potential is described elsewhere [7,66].

##### TRPM Subfamily

**TRPM2**—This channel is considered as an important sensor of oxidative stress [67], and is therefore likely to play a role in tumorigenesis. TRPM2 possesses a specific C-terminal domain responsible for binding ADP-ribose, which together with Ca^2+^ contributes to the channel’s activation [68]. Interestingly, TRPM2 expression was higher in disseminated circulating 4T1 tumor cells compared to the primary tumor, and its expression profile was associated with a more pronounced mesenchymal phenotype. The authors further provided evidence that increased TRPM2 expression is a consequence of EMT, rather than being involved in its induction, and that the reverse transition towards a more epithelial state has the opposite effect on TRPM2 [27]. Importantly, the increased TRPM2 expression that is associated with EMT in circulating tumor cells renders them more susceptible to neutrophil cytotoxicity, as neutrophil-secreted H_2_O_2_ will activate TRPM2 via ADP-ribose formation and result in calcium-overload associated apoptosis [67,69].

Functional expression of TRPM2 was also observed in gastric cancer. Additionally, silencing of TRPM2 in AGS cells, a human gastric adenocarcinoma cell-line, resulted in significant reductions in cell migration and invasion, accompanied by increased expression of E-cadherin and decreased levels of the EMT markers N-cadherin, snail, slug and MMPs. Moreover, TRPM2 silencing inhibited tumor formation by gastric cancer cells in a xenograft mouse model [28].

Silencing TRPM2 in a lung carcinoma cell line inhibited invasion and altered EMT marker gene expression. Moreover, similar results were observed in xenograft mouse models in which TRPM2 tissue expression was depleted [29]. Interestingly, introducing long non-coding TRPM2-anti-sense promotes EMT via SOX4 signaling in laryngeal squamous cell carcinoma cell lines, suggesting that the effect of TRPM2 on EMT induction in cancer cells might not be limited to its function as an ion channel [30]. Taken together, increased TRPM2 expression or functioning has been associated with EMT and consequently migration and invasion. 

**TRPM4**—In contrast to most of the TRP channels, TRPM4 is a Ca^2+^-activated, monovalent cation channel. Activation of the channel is facilitated by increased levels of cytosolic Ca^2+^, and induces an influx of monovalent cations, which causes membrane depolarization and reduces the driving force for Ca^2+^ entry via other ion channels [70]. TRPM4 has been shown to play an oncogenic role in prostate cancer. Overexpression is linked to increased recurrence risk [71]. Moreover, PC3 prostate cancer cells express 10-fold more TRPM4 mRNA than normal prostate epithelial cells. Silencing TRPM4 using shRNA resulted in increased β-catenin degradation, followed by decreased expression of β-catenin target genes and impaired cellular proliferation. In contrast, increased levels of TRPM4 in cancer will stabilize β-catenin, which is mediated by alterations in the calcium/calmodulin/Akt1 pathway [31]. Further research showed that decreasing TRPM4 expression by shRNA in PC3 prostate cancer cells resulted in decreased migration and invasion capabilities, along with reduced expression of Snail1. Together, these findings suggest that TRPM4 silencing indirectly reverses EMT process in these cells [32]. Relatedly, downregulation of TRPM4 by the administration of microRNA-150 also resulted in the inhibition of the EMT in prostate cancer cells. Collectively, the results suggest that silencing of TRPM4 by microRNA-150 or siRNA counteracts the EMT process and results in the suppression of proliferation, migration and invasion; and the promotion of apoptosis, thereby restraining tumorigenesis and metastasis in PC3 cells [33].

Recently, a similar role of TRPM4 in the regulation of the EMT process was reported in regard to breast cancer. These results showed that elevated TRPM4 levels correlated with later cancer stages, increased expression of EMT genes and increased expression of estrogen responsive genes [34]. 

In contrast to what is reported on prostate and breast cancer, decreased TRPM4 expression in endometrial cancer correlates with increased EMT and an unfavorable prognosis. Moreover, TRPM4 silencing in endometrial cancer cells increased expression of EMT markers, possibly via increased signaling of the PI3K/AKT/mTOR pathway [35]. 

**TRPM7**—TRPM7 is a ubiquitously expressed non-selective cation channel that includes a C-terminal kinase capable of phosphorylating downstream substrates. Moreover, TRPM7 plays an important role in Mg^2+^ homeostasis in various cell types [72]. In human breast cancer cells (MDA-MB-458 cells), EGF induced a calcium transient that was accompanied by increased vimentin expression, which is suggestive of EMT. Moreover, intracellular calcium chelation prevented EGF-induced increases of EMT-associated genes, including N-cadherin and Twist. Important to note is that increased cytosolic calcium signaling alone is not sufficient to induce EMT, as calcium mobilizing agents trypsin and ATP did not induce an EMT-like phenotype. These findings highlight again that the specific spatial and temporal aspects of the calcium signal regulate calcium-dependent EMT [2]. Blocking TRPM7 expression with siRNA, or its function using antagonist NS8593, decreased EGF-induced vimentin induction on the protein level, but not on the mRNA level. Interestingly, this effect was independent of an EGF-induced cytosolic calcium increase. Moreover, TRPM7 was not directly activated by EGF but might play a role downstream to EGFR in ERK1/2 and STAT3 activation. However, silencing of TRPM7 did not prevent EGF-induced calcium-dependent upregulation of EMT markers, suggesting the involvement other calcium transporters [2]. In two other representative cell lines of breast cancer, it was demonstrated that TRPM7 regulates mesenchymal features of these cells by regulation of SOX4 expression. Interestingly, TRPM7-induced cytoskeletal relaxation promoted SOX4 expression, thereby instigating the acquisition of mesenchymal characteristics [36]. Additionally, it was shown by Vanlaeys and colleagues that cadmium exerts its carcinogenic effect, i.e. EMT induction, via TRPM7 activation in healthy breast epithelial cells [73].

In ovarian cancer, TRPM7 expression was significantly increased compared to non-tumor tissues, and was negatively correlated with E-cadherin and positively correlated with mesenchymal markers vimentin and Twist. Moreover, TRPM7 silencing in ovarian cancer cell lines SKOV3 and OVCAR3 reduced EGF-dependent migration, invasion and wound healing. This was accompanied by increased levels of E-cadherin and decreased levels of N-cadherin, vimentin and Twist. The authors further suggested that decreased levels of cytosolic calcium by silencing TRPM7 might impair EMT through prevention of PI3K/AKT activation [37].

In prostate cancer cells PC3 and DU145, hypoxia-induced EMT resulted in increased levels of TRPM7, vimentin and N-cadherin, and decreased levels of E-cadherin, implying that TRPM7 expression was correlated with EMT. Moreover, silencing of TRPM7 mitigated hypoxia-induced EMT by promoting degradation of HIF-1α in the proteasome [39]. In addition, knockdown of TRPM7 in PC3 cells reversed EMT, resulting in downregulation of MMPs and upregulation of E-cadherin, thereby inhibiting migration and invasion [38].

Finally, it was reported that TRPM7 expression promotes migratory and metastatic characteristics via regulation of gene expression profiles in neuroblastoma cell lines involving the EMT transcription factor SNAI2 [40]. Taken together, these reports suggest that TRPM7 is associated with EMT. 

**TRPM8**—In humans, TRPM8 is the primary molecular transducer for indicating coldness, and is mainly expressed in sensory nerve endings [74,75]. Interestingly, it was shown that overexpression of TRPM8 induced EMT in breast cancer cells by upregulating vimentin and fibronectin while downregulating E-cadherin. TRPM8 expression was upregulated in breast cancer cell samples. This increased expression would promote EMT by activating the PI3K/Akt pathway, and this might result in increased migration and invasion [41]. In contrast, TRPM8 expression has been shown to exert an anti-invasive effect on prostate cancer cells, contradicting an EMT-promoting role of the channel [76].

##### TRPC Subfamily

Members of the TRPC subfamily form heteromultimers within the confines of TRPC3/6/7, which are activated by phospholipase C-generated diacylglycerols (DAGs), and TRPC1/4/5, of which TRPC1 is activated by STIM1 [77,78,79]. 

**TRPC1**—TGFβ-induced EMT was shown to be mediated by store operated calcium entry (SOCE) involving STIM1 and TRPC1 in breast cancer cells. The authors stated that this calcium influx may be necessary for activation of calpains and MMPs, which in turn could regulate cellular adherence and promote migration [43]. Accordingly, Azimi and colleagues reported increased TRPC1 expression after hypoxia-induced EMT via HIF-1α signaling in breast cancer cells. Furthermore, reducing TRPC1 expression inhibited the hypoxia-induced expression of snail, vimentin and Twist [42]. Additionally, it was also demonstrated that the claudin-low breast cancer subtype exhibited the highest TRPC1 expression levels compared to other subtypes. Moreover, in triple negative breast cancer, the mesenchymal subtype showed the highest expression level of TRPC1. The basal subtype with lymph-node metastasis was associated with high TRPC1 expression and worse prognosis [42].

In gastric cancer cells, a role for the TRPC1/3/6 in TGFβ-induced EMT was described. Notably, activation of the Ras/Raf1/ERK1/2 signaling cascade was suppressed by the TRPC1/3/6 inhibitors SKF96365 and 2-APB [44].

TGFβ-induced calcium entry via TRPC1 was also reported to induce invasion in pancreatic cancer cells. Indeed, TRPC1 expression and functioning was required to activate PKCa in response to TGFβ in these cells [45]. Hence, these reports are in line with the observation that TRPC1 has been associated with the mesenchymal phenotype.

**TRPC3**—As described above, a role for TRPC3 in TGFβ-induced EMT was described in gastric cancer cells, one mediated via the Ras/Raf1/ERK1/2 signaling cascade [44]. 

**TRPC5**—In colon cancer, TRPC5 was correlated with tumor metastasis, likely by mediating HIF-1α expression and thereby activating Twist and the induction of EMT [48].

**TRPC6**—This channel is highly expressed in hepatic carcinoma samples. This increased calcium influx through TRPC6 could mediate sustained calcium accumulation that potentially plays a role in inducing EMT, HIF-1α signaling and DNA repair to confer multi-drug resistance to the cells [49]. Together with TRPC1 and TRPC3, a role for TRPC6 in TGFβ-induced EMT was described in gastric cancer cells, mediated via the Ras/Raf1/ERK1/2 signaling cascade [44].

##### TRPV Subfamily

**TRPV2**—The TRPV2 channel, formerly known as the growth-factor regulated channel, was initially characterized as a noxious heat sensor, but more recently suggested to have roles in osmosensory and mechanosensory mechanisms, and cellular motility [80,81,82]. Expression and proper functioning of TRPV2 have been associated with more pronounced cellular migration and invasion in various cancer cell types [83,84,85,86]. However, a direct link between altered TRPV2 activity and EMT induction has not been clearly described. Recently, pathway analysis conducted in esophageal squamous cell carcinoma revealed roles for TRPV2, WNT/β-catenin signaling and EMT. Indeed, depletion of TRPV2 in esophageal squamous cell carcinoma cell lines resulted in decreased expression of WNT10A, TGFβ2, TGFβR2, GLI, Snai1, Zeb2, CDH2, CD44 and SOX2, and decreased the migratory potential [50].

**TRPV4**—TRPV4 expression is well described for various epithelial tissues and is activated by warm temperatures, mechanical forces and lipid mediators such as arachidonic acid and its metabolites [9,10,87,88]. A recent review focused on the potential role of TRPV4 in tumorigenesis and its therapeutic potential [89]. Abnormal expression of TRPV4 was shown to be related to tumor formation and metastasis. The latter is related to the strong link between TRPV4 and cytoskeletal proteins actin and E-cadherin, two factors that are crucial features of the EMT. In normal keratinocytes, both TGFβ1-induced and matrix stiffness-induced EMT were impaired when TRPV4 was inhibited. Moreover, TRPV4 activity mediates TAZ, but not YAP, expression and nuclear accumulation, a process that promotes matrix stiffness-induced EMT [51,52]. Interestingly, in hepatocellular carcinoma cells, inhibition of TRPV4 with HC-067047 suppressed migration by repression of the ERK pathway [53].

**TRPV6**—This ion channel is primarily expressed in epithelial cells and is, along with TRPV5, a major player in Ca^2+^ reabsorption in the intestine [90]. Recently, it was demonstrated that TRPV6 is involved in maintaining mammary epithelial cells’ integrity via its connections to E-cadherin and the cytoskeleton. Indeed, loss of TRPV6 expression in these cells resulted in disruption of cell–cell interactions and abnormal 3D-mammo sphere morphology. Moreover, TRPV6 expression was associated with increased levels of EMT markers, suggesting that either too low or too high expression of the channel can lead to the development of breast cancer [54]. Indeed, aberrant TRPV6 expression has been detected in several cancer types, including breast cancer and pancreatic cancer [91,92]. The potential value of TRPV6 as a therapeutic target in cancer treatment was recently reviewed [93]. 

##### TRPP Subfamily

Members of the TRPP subfamily are mechanosensitive non-selective cation channels that are associated with the development of polycystic kidney disease (PKD) [94]. Polycystin-1 and 2 (PKD 1/2) are the most studied members of this subfamily. They interact to allow cation influx into the cell, and are primarily expressed in renal primary cilia. EMT is suggested to be involved in the pathophysiology of PKD, which is indicative of a role for PKD channels in this transition process [95].

Indeed, disruption of the connection between PKD1 and E-cadherin due to increased phosphorylation of PKD1 leads to dedifferentiated PKD cells, characterized by an increased expression of mesenchymal N-cadherin [96]. Moreover, Boca and colleagues described how PKD1 could promote cell scattering and migration via actin cytoskeleton reorganization in renal epithelial cells trough PI3K signaling [55].

In colorectal cancer, it was demonstrated that both PKD1 and PKD2 promote EMT in cell lines and xenografts, and are predictors for reduced recurrence-free survival and overall survival [56]. 

#### 2.2.2. Fibrosis

Wound healing after tissue damage is driven by an acute inflammatory response in which various immune cells are recruited towards the injured site and start secreting various cytokines and GFs, including TGFβ1. In normal conditions, this leads to an EMT reaction, and consequently the migration of fibroblasts to the injured site in order to initiate the repairing process. However, fibrosis is caused by chronic inflammation and secretion of cytokines that will induce prolonged EMT, resulting in persistent differentiation of fibroblasts into myofibroblasts. These myofibroblasts secrete products that interfere with the normal extracellular matrix, causing tissue remodeling and damage, followed by the deposition of connective tissue and the formation of permanent scar tissue. 

Several TRP channels have been linked to myofibroblast differentiation and fibrosis, as recently reviewed by Inoue and colleagues [97]. For instance, in patients with atrial fibrillation, a pathology caused by cardiac fibrosis, TRPM7 expression and currents were significantly elevated in fibroblasts. Moreover, these fibroblasts were remarkably more prone to myofibroblast differentiation, which could be diminished by knockdown of TRPM7 [98].

TRPA1 is involved in TGFβ1-dependent fibrosis in the mouse’s corneal stroma [99]. Indeed, loss of TRPA1 or its pharmacological inhibition in mice reduced inflammation and fibrosis via the inhibition of TGFβ1 signaling.

In lung fibroblasts, TRPV4 mediates matrix-stiffness-dependent myofibroblast differentiation [100]. A more recent study confirmed these findings and suggested that the pro-fibrotic effect of TRPV4 activation was regulated via PI3Kγ translocation [101].

Pharmacological inhibition and knockdown of TRPC3 in renal fibroblasts inhibited myofibroblast differentiation by suppressing phosphorylation of ERK1/2, suggesting a role for the channel in the development of renal fibrosis [102].

A more detailed overview of TRP channels in fibrosis pathophysiology is described elsewhere [97,103].

#### 2.2.3. Other Pathologies

Chronic obstructive pulmonary disease (COPD) is a disease associated with significant EMT in airway epithelial cells. Higher levels of the mesenchymal marker vimentin were observed in lung tissues of COPD patients and were associated with increased TRPC1 expression. Furthermore, the application of cigarette smoke extract to human bronchial epithelial cells (16HBE) resulted in the induction of EMT. This was evidenced by increased vimentin expression and decreased E-cadherin expression, a phenomenon that was exacerbated by overexpression of TRPC1 [47]. Recently, a similar finding was observed by subjecting 16HBE cells to cyclic mechanical stretching to induce EMT, followed by both increased molecular and functional expression of TRPC1. As such, a robust increase in intracellular calcium was observed after cell exposure to stretching, which could be attenuated by silencing TRPC1. These findings showed that stretching of 16HBE cells induced TRPC1-dependent calcium influx, which caused the induction of EMT [46].

## 3. Conclusion and Perspectives

In recent years, TRP channels have been closely investigated as potential key players in EMT, especially due to their contributions to cancer progression and dissemination. Indeed, altered TRP channel expression and functioning have been observed in various cancer types. While their ability to sense and translate micro-environmental changes into cellular behavior is essential for various physiological processes, altered TRP channel expression patterns and consequently altered cytosolic calcium and sodium signaling could severely disrupt cellular functioning. Calcium is an important secondary messenger and a key player in various cellular processes, such as proliferation and cellular motility. Therefore, alterations in intracellular calcium concentrations mediated by alterations in TRP channel expression or functioning will result in altered cellular behavior. Those modifications in expression and functioning can be the results of aberrant extracellular signaling, or can be direct consequences of the cellular phenotype switch. Figure 2 summarizes various modes of TRP channel interference with cellular signaling cascades to induce EMT. Since EMT drives the most lethal characteristics of cancer, such as metastasis and the development of chemo resistance, TRP channels represent an attractive target in oncology. Direct targeting of EMT effector molecules remains pharmacologically challenging, given, for example, their many protein–protein interactions, as reviewed by Lazo et al. [104]. Therefore, targeting proteins that contribute to the induction of the transition might be more attractive. As stated here in this review, TRP channel-mediated calcium influx is an upstream event during EMT in several different cancer cell types. Moreover, the altered TRP channel expression in tumors, together with a strong presence of TRP channels at the cell surface, renders them interesting candidates for pharmacological modulation. Therefore, TRP channels might represent a new group of oncological targets to prevent cancer progression and metastasis. Finally, we speculate that TRP channels might also be involved in regulating EMT in non-malignant physiological processes, such as trophoblast invasion, embryonic development, wound healing and fibrosis, further increasing their potential as therapeutic targets.

## Figures and Tables

**Figure 1 ijms-22-08188-f001:**
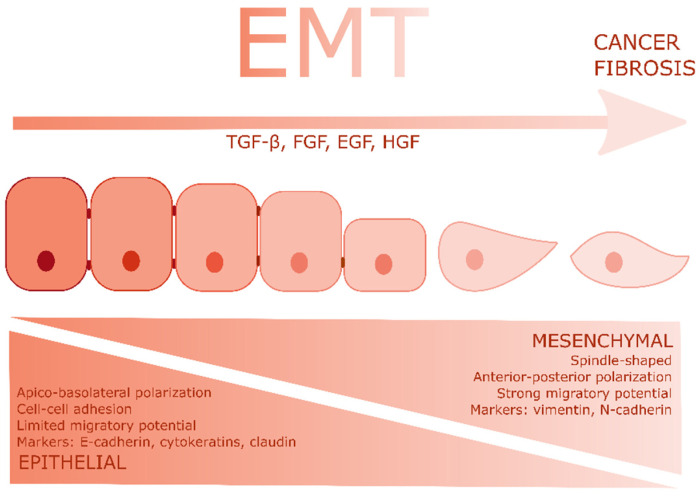
Epithelial to mesenchymal transition: TGFβ and growth factors induce EMT in epithelial cells, which is accompanied by the loss of typical epithelial features and the acquisition of mesenchymal characteristics.

**Figure 2 ijms-22-08188-f002:**
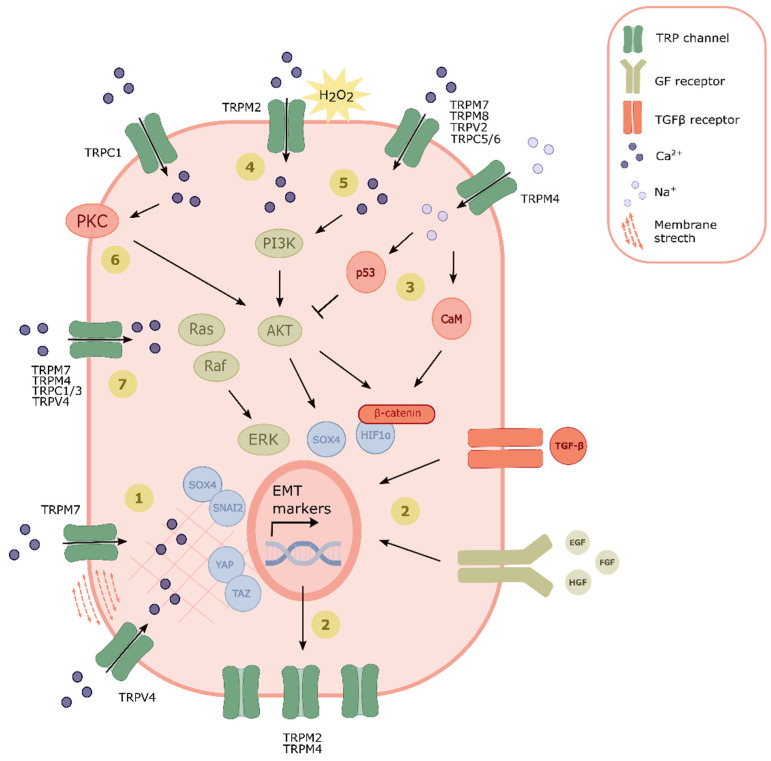
TRP channel signaling in EMT: (1) Membrane stretching caused by extracellular stimuli or cytoskeletal alterations induces TRP channel activation and promotes SOX4/SNAI2 and/or YAP/TAZ signaling. (2) TGFB and GF bind to their respective receptors and induce EMT, thereby upregulating TRP channel expression. (3) Entry of Na^+^ via TRPM4 promotes p53, thereby preventing EMT induction via inhibiting AKT signaling. Alternatively, calmodulin (CaM) activation via TRPM4 activity leads to β-catenin/HIF-1α stabilization and EMT induction. (4) ROS-induced TRP channel activation promotes PI3K/AKT pathway signaling, promoting EMT induction. (5) Other TRP channels promoting EMT via the PI3K/AKT pathway are TRPM7, TRPM8, TRPV2, TRPC5 and TRPC6. (6) Activation of PKC and AKT via TRP channel mediated Ca^2+^ entry (7) Activation of the Ras/Raf/ERK pathway via TRP channel signaling promotes EMT induction.

**Table 1 ijms-22-08188-t001:** An overview of TRP channels in the EMT.

Channel	Observation	Cell Line/Cancer Type	Mechanism/Affected Target	References
TRPM2	Increased expression	4T1 breast cancer cell line/HMLE human mammary tumor cells	Increased TRPM2 expression and susceptibility towards H_2_O_2_	[27]
	Promotes EMT	AGS gastric cancer cell line	Promotes EMT via AKT stimulation	[28]
		A549 and H1299 lung cancer cell lines	/	[29]
		Laryngeal squamous cell carcinoma cells	Promotes SOX-4 signaling	[30]
TRPM4	Promotes EMT	PC3 and LnCaP prostate cancer cell lines	Stabilization of β-catenin via calmodulin/AKT1 pathway	[31,32,33]
	Increased expression	Breast cancer tissue	/	[34]
	Inhibits EMT	Endometrial cancer cell lines (AN3CA, ishikawa, HEC-1A, HEC-1B, RL-95, primary cell lines)	Inhibition of PI3K/AKT/mTOR signaling via P53	[35]
TRPM7	Promotes EMT	Breast cancer cell lines MDA-MB-485, MDA-MB-231 and Hs 578T	EGF-induced VIM induction via ERK/STAT signaling/SOX4 regulation via cytoskeletal relaxation	[2,36]
		Ovarian cancer cell lines SKOV3 and OVCAR3	PI3K/AKT signaling	[37]
		Prostate cancer cell lines PC3 and DU145	Hypoxia-induced EMT via HIF-1α signaling	[38,39]
		Neuroblastoma cell lines: N1E-115, SH-EP2 and SH-SY5Y	SNAI2 induction	[40]
TRPM8	Promotes EMT	Breast cancer cell lines MCF-7, T47D, MDA-MB-231, BT549, SKBR3 and ZR-75-30	PI3K/Akt signaling	[41]
TRPC1	Promotes EMT	Healthy epithelial breast cell line NMuMG + breast cancer cell lines MCF-10A, MDA-MB-231, MDA-MB-468 and HCC1569 + breast cancer tissue	Activation of calpains and MMPs via SOCE/HIF-1α signaling	[42,43]
		Gastric cancer cell line SGC-7901 cells	Ras/Raf1/ERK1/2 signaling	[44]
		Pancreatic cell line BxPc3	PKCα activation	[45]
		Lung epithelial cell line 16HBE	/	[46,47]
TRPC3	Promotes EMT	Gastric cancer cell line SGC-7901 cells	Ras/Raf1/ERK1/2 signaling	[44]
TRPC5	Promotes EMT	Colon cancer cell lines SW620, RKO, SW1116, HT29, and HCT116	HIF-1α signaling	[48]
TRPC6	Promotes EMT	Hepatocellular carcinoma cell lines Huh7 and HepG2	HIF-1α signaling	[49]
		Gastric cancer cell line SGC-7901 cells	Ras/Raf1/ERK1/2 signaling	[44]
TRPV2	Promotes EMT	Esophageal squamous cell carcinoma	WNT/β-catenin signaling	[50]
TRPV4	Promotes EMT	Keratinocytes	Matrix stiffness induced EMT via TAZ signaling	[51,52]
		Hepatocellular carcinoma cell lines LO2, Huh7 and HepG2	ERK signaling pathway	[53]
TRPV6	Prevents EMT	Breast cancer cell lines MCF-10A, 184A1 and MDA-MB231	Maintains epithelial integrity via E-cadherin and cytoskeletal connection (CaM-dependent kinases)	[54]
	Promotes EMT		Upregulation of EMT markers via PI3K/AKT signaling	
TRPP1 (PKD1)	Promotes EMT	Renal epithelial (MDCK) cells	Cytoskeletal reorganisation via the PI3K pathway	[55]
	Promotes EMT	Colorectal cancer cells HCT116	/	[56]
TRPP2 (PKD2)	Promotes EMT	Colorectal cancer cells SW480	mTOR pathway	[56]

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
