# Peer review of "Transient Receptor Potential Channels in the Epithelial-to-Mesenchymal Transition"

_ijms, 2021, doi:10.3390/ijms22158188_

Round 1
Reviewer 1 Report
A review article of Eynde et al. Transient Receptor Potential channels in Epithelial-to-Mesenchymal transition provides an excellent update for an impotent research area. The manuscript is well focused and excellently written. I have no critical points to suggest.
Author Response
Dear Reviewer,
We would like to express our sincere gratitude for the positive feedback.
Reviewer 2 Report
In this review, the authors describe the role of TRP channels in EMT. After the reading, the main role of TRP channels is described in cancer, with a tiny paragraph of their roles in fibrosis.
TRP channels have been well studied since their discovery in in the beginning of the 90’s. Thus, several sub-families exist, but in this review the authors only focused on TRPM, TRPC and TRPV sub-families… what about TRPA, TRPP and TRPML ? It means that this review is uncomplete or the authors must give an explanation.
Furthermore, inside sub-families, it is also uncomplete, for example, in TRPC subfamily, only C1, C5 and C6 are referenced. But the role of C3 in renal fibrosis (Saliba 2015), and in migration/invasion of ovarian cancer cells (Yang 2009) is documented.
In 30 years, the TRP channels have been characterized and their ability to transport Ca2+ ions (or not) is established. Thus, the complete review on TRP channels made by Bernd Nilius’ group (Gees et al., 2010), showed that TRP channels have very different selectivity, TRPM4 is not able to transport Ca2+ ions, when TRPV5 and V6 are real Ca2+ channels with a Pca/Pna ratio of 100. It means that these channels are more cationic channels than calcium’s. Even if they can transport Ca2+ ions, in physiological conditions, these channels transport more Na+ ions than Ca2+ ones. It is therefore difficult to read that these channels are calcium-permeable and modulate the intracellular calcium concentration (lines 18 to 21)… Thus, TRP channels can depolarized the plasma membrane due to Na+ fluxes, inducing a decrease of the Ca2+ driving force, and can also activate Ca2+ voltage-gated channels…
The table 1 is useless, because the same experiments and conclusions are described in the text.
Fibrosis is also a pathology where the role of TRP channels have been described, but the authors should develop it.
Author Response
In this review, the authors describe the role of TRP channels in EMT. After the reading, the main role of TRP channels is described in cancer, with a tiny paragraph of their roles in fibrosis.
We would like to clarify that the scope of this review was to summarize the involvement of TRP channels in phenotypic switching of cell types during the EMT process, both physiological and pathophysiological (cancer, fibrosis). It was not our purpose to provide a comprehensive overview of TRP channel function in said pathologies as such, as they have been excellently reviewed before. These specific references were included in the revised manuscript (Chen et al 2014, Santoni et al. 2011, Inoue et al. 2019, Yue et al. 2013).
TRP channels have been well studied since their discovery in in the beginning of the 90’s. Thus, several sub-families exist, but in this review the authors only focused on TRPM, TRPC and TRPV sub-families… what about TRPA, TRPP and TRPML? It means that this review is uncomplete or the authors must give an explanation.
We thank the reviewer for these remarks and would like to provide an explanation for the exclusion of certain sub-families.
- For TRPA channels we could not find any evidence for its involvement in the EMT process and are therefore not included in the review.
- Although TRPML channels have recently been described in cancer (Santoni 2020 and Xu 2021), no direct connections to the EMT process were described yet. Therefore, we did not include this subfamily in this overview.
We acknowledge the incorrect exclusion of TRPP channels in our review and included a section to the revised manuscript where we elaborate on this topic (line 438-453)
Furthermore, inside sub-families, it is also uncomplete, for example, in TRPC subfamily, only C1, C5 and C6 are referenced. But the role of C3 in renal fibrosis (Saliba 2015), and in migration/invasion of ovarian cancer cells (Yang 2009) is documented.
We appreciate this comment and would like to clarify these concerns. A role for TRPC3 in EMT in cancer is described in combination with TRPC1 and TRPC6, and can be found under the TRPC1 paragraph. In addition, a separate paragraph (and table section) has been included in the revised manuscript regarding the role of TRPC3 in both the cancer and fibrosis (line 387-388 and line 477-479). As the scope of the review was to included only those TRP channel for which a clear role during EMT has been described, we chose not to refer to Yang 2009. Although Yang et al describe an effect of TRPC3 on cancer cell proliferation, direct evidence for the involvement in the EMT process was missing. Therefore, this paper has not been included in the revised manuscript.
In 30 years, the TRP channels have been characterized and their ability to transport Ca2+ ions (or not) is established. Thus, the complete review on TRP channels made by Bernd Nilius’ group (Gees et al., 2010), showed that TRP channels have very different selectivity, TRPM4 is not able to transport Ca2+ ions, when TRPV5 and V6 are real Ca2+ channels with a Pca/Pna ratio of 100. It means that these channels are more cationic channels than calcium’s. Even if they can transport Ca2+ ions, in physiological conditions, these channels transport more Na+ ions than Ca2+ ones. It is therefore difficult to read that these channels are calcium-permeable and modulate the intracellular calcium concentration (lines 18 to 21)… Thus, TRP channels can depolarized the plasma membrane due to Na+ fluxes, inducing a decrease of the Ca2+ driving force, and can also activate Ca2+ voltage-gated channels…
We thank the reviewer for these remarks and adjusted the text accordingly. As such, the reference of TRP channels as calcium-permeable channels was adapted to non-selective cation channels. In additional, the cation selectivity of TRP channels was concisely added to the revised manuscript (line 93-95).
The table 1 is useless, because the same experiments and conclusions are described in the text.
We appreciate the reviewers comment but would like to argue that this table is intended as a schematic overview of what is written in the text, providing a quick and concise tool to interpret all the information. Therefore, we would like to include the table in this manuscript but it might be moved it the supplementary section if this is preferred.
Fibrosis is also a pathology where the role of TRP channels have been described, but the authors should develop it.
As stated by the reviewers, TRP channels have been described in fibrosis. However, research about the role of TRP channels in fibrosis is a relatively new field, and not many TRP channels are described in the development of this pathology yet. Most of what is known was already described in the text, and we added some additional references. As the scope of this review as to focus on the role of TRP channels in the EMT process itself, a comprehensive overview of TRP channels function in fibrosis, without a clear link to EMT, was excluded.
In this review, the authors describe the role of TRP channels in EMT. After the reading, the main role of TRP channels is described in cancer, with a tiny paragraph of their roles in fibrosis.
We would like to clarify that the scope of this review was to summarize the involvement of TRP channels in phenotypic switching of cell types during the EMT process, both physiological and pathophysiological (cancer, fibrosis). It was not our purpose to provide a comprehensive overview of TRP channel function in said pathologies as such, as they have been excellently reviewed before. These specific references were included in the revised manuscript (Chen et al 2014, Santoni et al. 2011, Inoue et al. 2019, Yue et al. 2013).
TRP channels have been well studied since their discovery in in the beginning of the 90’s. Thus, several sub-families exist, but in this review the authors only focused on TRPM, TRPC and TRPV sub-families… what about TRPA, TRPP and TRPML? It means that this review is uncomplete or the authors must give an explanation.
We thank the reviewer for these remarks and would like to provide an explanation for the exclusion of certain sub-families.
- For TRPA channels we could not find any evidence for its involvement in the EMT process and are therefore not included in the review.
- Although TRPML channels have recently been described in cancer (Santoni 2020 and Xu 2021), no direct connections to the EMT process were described yet. Therefore, we did not include this subfamily in this overview.
We acknowledge the incorrect exclusion of TRPP channels in our review and included a section to the revised manuscript where we elaborate on this topic (line 438-453)
Furthermore, inside sub-families, it is also uncomplete, for example, in TRPC subfamily, only C1, C5 and C6 are referenced. But the role of C3 in renal fibrosis (Saliba 2015), and in migration/invasion of ovarian cancer cells (Yang 2009) is documented.
We appreciate this comment and would like to clarify these concerns. A role for TRPC3 in EMT in cancer is described in combination with TRPC1 and TRPC6, and can be found under the TRPC1 paragraph. In addition, a separate paragraph (and table section) has been included in the revised manuscript regarding the role of TRPC3 in both the cancer and fibrosis (line 387-388 and line 477-479). As the scope of the review was to included only those TRP channel for which a clear role during EMT has been described, we chose not to refer to Yang 2009. Although Yang et al describe an effect of TRPC3 on cancer cell proliferation, direct evidence for the involvement in the EMT process was missing. Therefore, this paper has not been included in the revised manuscript.
In 30 years, the TRP channels have been characterized and their ability to transport Ca2+ ions (or not) is established. Thus, the complete review on TRP channels made by Bernd Nilius’ group (Gees et al., 2010), showed that TRP channels have very different selectivity, TRPM4 is not able to transport Ca2+ ions, when TRPV5 and V6 are real Ca2+ channels with a Pca/Pna ratio of 100. It means that these channels are more cationic channels than calcium’s. Even if they can transport Ca2+ ions, in physiological conditions, these channels transport more Na+ ions than Ca2+ ones. It is therefore difficult to read that these channels are calcium-permeable and modulate the intracellular calcium concentration (lines 18 to 21)… Thus, TRP channels can depolarized the plasma membrane due to Na+ fluxes, inducing a decrease of the Ca2+ driving force, and can also activate Ca2+ voltage-gated channels…
We thank the reviewer for these remarks and adjusted the text accordingly. As such, the reference of TRP channels as calcium-permeable channels was adapted to non-selective cation channels. In additional, the cation selectivity of TRP channels was concisely added to the revised manuscript (line 93-95).
The table 1 is useless, because the same experiments and conclusions are described in the text.
We appreciate the reviewers comment but would like to argue that this table is intended as a schematic overview of what is written in the text, providing a quick and concise tool to interpret all the information. Therefore, we would like to include the table in this manuscript but it might be moved it the supplementary section if this is preferred.
Fibrosis is also a pathology where the role of TRP channels have been described, but the authors should develop it.
As stated by the reviewers, TRP channels have been described in fibrosis. However, research about the role of TRP channels in fibrosis is a relatively new field, and not many TRP channels are described in the development of this pathology yet. Most of what is known was already described in the text, and we added some additional references. As the scope of this review as to focus on the role of TRP channels in the EMT process itself, a comprehensive overview of TRP channels function in fibrosis, without a clear link to EMT, was excluded.
